# Investigation of the Pharmacodynamic Components of *Gastrodia elata* Blume for Treatment of Type 2 Diabetes Mellitus through HPLC, Bioactivity, Network Pharmacology and Molecular Docking

**DOI:** 10.3390/ijms251910498

**Published:** 2024-09-29

**Authors:** Xiu Yang, Lilang Li, Yanfang Yan, Xuehao Hu, Qiji Li, Liangqun Li, Yu Wang, Xian Tao, Lishou Yang, Mei Peng, Juan Yang, Xiaosheng Yang, Ming Gao

**Affiliations:** 1State Key Laboratory for Functions and Applications of Medicinal Plants, Guizhou Medical University, Guiyang 550014, China; yangxiu058@gmail.com (X.Y.); lililang@gmc.edu.cn (L.L.); yangfangyan1926@sina.com (Y.Y.); 15720531029@163.com (X.H.); leeqiji@126.com (Q.L.); liliangqun2010@163.com (L.L.); wang_yu@gmc.edu.cn (Y.W.); 15761402947@163.com (X.T.); lishouyang2023@sina.cn (L.Y.); pengmei520@163.com (M.P.); yangxz2002@126.com (J.Y.); 2Natural Products Research Center of Guizhou Province, Guiyang 550014, China; 3College of Pharmacy, Guizhou University of Traditional Chinese Medicine, Guiyang 550014, China

**Keywords:** *Gastrodia elata* Blume, type 2 diabetes mellitus (T2DM), α-glucosidase, antioxidant, network pharmacology, molecular docking

## Abstract

The occurrence of type 2 diabetes mellitus (T2DM), a worldwide chronic disease, is mainly caused by insufficient insulin production and places a huge burden on the health system. *Gastrodia elata* Blume (GE), a food of medicine–food homology, has been reported to have the ability to inhibit glycosidase activity, indicating its potential in the treatment of diabetes. However, the main pharmacological components of GE for the treatment of T2DM have not been fully clarified. Therefore, this study aims to clarify the pharmacological components changes of GE with different drying methods and the treatment of T2DM using HPLC, network pharmacology, molecular docking and experimental evaluations. The results showed that the GE samples processed by the steam-lyophilized method possessed the highest total content of the six marker components and the strongest antioxidant and α-glucosidase inhibitory abilities. Meanwhile, the six marker compounds had a total of 238 T2DM-related gene targets. Notably, these active compounds have good affinity for key gene targets associated with T2DM signaling pathways. In conclusion, this study revealed that different drying methods of GE affect the content of its major active compounds, antioxidant capacity, α-glucosidase inhibitory capacity and potential pharmacological effects on T2DM, indicating that it is a potential treatment of T2DM.

## 1. Introduction

Type 2 diabetes mellitus (T2DM) is the most common chronic form of diabetes, with T2DM accounting for more than 90% of all diabetic patients and characterized by resistance to insulin action and an inadequate compensatory insulin secretory response [1]. This condition leads to the prolonged and progressive development of hyperglycemia, in which the negative feedback loop between insulin activity and insulin secretion becomes dysregulated, resulting in abnormal glucose metabolism [2]. What is worse, the various tissues and organs in diabetes patients could been damaged due to the long-term instability of blood sugar, leading to the occurrence of complications, such as coronary artery disease, conduction disturbances, infectious complications, etc. [3,4]. Oxidative stress (OS) induced by hyperglycemia triggers the release of pro-inflammatory cytokines from the cells of the innate immune system, leading to the overproduction of free radicals through the activation of macrophages [5]. The OS, advanced glycosylation end products, and inflammation are inextricably intertwined in the context of diabetes and related complications [6]. The inhibition of glycohydrolases is a commonly used therapeutic strategy for controlling hyperglycemia, such as α-glucosidase and α-amylase, to prevent the breakdown of complex dietary carbohydrates into simple monosaccharides and their subsequent absorption in the small intestine [7]. Currently, the mainstay of treatment for T2DM consists of insulin and a variety of oral hypoglycemic agents such as α-glucosidase inhibitors, sulfonylureas, metformin and thiazolidinediones [8]. However, these drugs have significant side effects, with hypoglycemia, weight gain and increased gastrointestinal problems [9]. Therefore, the targeted discovery of the α-glucosidase inhibitor and OS modulator for the treatment of diabetic patients from medicine–food homology plants with better safety and low toxicity could be an effective approach [10].

*Gastrodia elata* Blume (GE) is a valuable traditional medicinal material that has been used for the treatment of convulsions, epilepsy, tetanus, vertigo, paralysis, etc. [11]. In 2019, the Chinese government officially announced that GE can be used as both a medicine and food based on a long history of folk food. Moreover, modern pharmacological studies have verified that the extracts of GE possess neuroprotective, antioxidant, anxiolytic, hypoglycemic, antidepressant, sedative and anti-inflammatory activities [12]. Recently, studies have found that GE and its active components have significant antidiabetic effects by regulating hypothalamic insulin signaling and promoting pancreatic β-cell proliferation. Among them, gastrodian (GAS), a main ingredient of GE, showed a potential antidiabetic effect via the improvement in β-cell function, the enhancement of insulin signaling, the reduction in insulin resistance as well as the alleviation of many diabetic complications through anti-inflammatory and antioxidant properties and the inhibition of endoplasmic reticulum stress [13].

At the present stage, GAS and p-hydroxybenzyl alcohol (HBA) are listed in the Chinese Pharmacopoeia as indicators of the quality evaluation of GE. Meanwhile, studies have confirmed that parishins also play a vital role in the pharmacodynamics of GE with higher contents than GAS [14]. Moreover, the different drying methods for herbal medicine could lead to significant differences in the chemical composition of herbal medicine [15]. Therefore, the effects of different compound types on the quality of GE should be considered comprehensively in quality evaluation. A large number of studies have been conducted on the treatment effect of GE on central nervous system diseases, such as insomnia and depression, and few reports have been reported on the treatment of diabetes mellitus.

With the development of systems biology and computer technology, a novel strategy, network pharmacology, has been applied to systematically and comprehensively explore the intervention and influence of drugs on the disease network based on the “disease–gene–target–drug” interaction network, which could find effective substances in complex systems, predict the effective effects of known compounds, and provide valuable information for explaining the mechanism of action [16,17]. Thus, this study aimed to explore the changes in the content of six marker components, α-glucosidase inhibition and antioxidant activities of GE processed with different drying methods and systematically investigate the effects of the main components of GE on T2DM and its mechanisms using network pharmacology and molecular docking techniques.

## 2. Results

### 2.1. The Contents of Marker Components of GE by HPLC

#### 2.1.1. The Standard Curve of Marker Components of GE

The sample solution (10 µL) was injected into the HPLC system to analyze the six marker components and then compared with the mixed solution of standard chemicals based on the retention time of each peak in the chromatogram (Figure 1, Table 1). The RSDs of precision, stability, repeatability and accuracy were <3% (Appendix A), indicating that the method is feasible for the analysis of GE samples. Notably, the relationship values showed that the correlation between the peak area and the concentration of the six marker components was good, with an R^2^ > 0.9990, and the linear range was normal (Table 2, Appendix A).

#### 2.1.2. Analysis of the Content of Marker Components in GE Samples by HPLC

The GE sample solutions treated with different drying methods were injected into the HPLC system using the same method mentioned in Section 2.1.1. The contents of each marker component were confirmed by calculating the corresponding peak area. The results indicated that the total contents of the six marker components in the wild-imitated cultivation of GE were higher than the bag-planted ones under the same drying method (Table 3). The contents of HBA and parishin C in the GE samples treated by FD, FAD and FFD were higher than those in the GE samples with SD, SAD, SFD and the SF9 drying method (Table 4). The combined application of steaming and drying methods can result in an increase in the contents of GAS, parishin E, parishin B and parishin A, which were significantly higher than that of fresh GE. Differences in the *Gastrodia elata* Blume composition of the batch plots are shown in Appendix A.

### 2.2. Antioxidant Activities and α-Glucosidase Inhibition Assay

#### 2.2.1. Antioxidant Activities

The scavenging capacities of one batch of GE samples treated by seven different drying methods were determined against DPPH, ABTS and OH· to adequately evaluate the antioxidant activity of GE. The three antioxidant capacity tests of GE samples showed a significant dose-dependence over the concentration range, with the better scavenging of DPPH, ABTS and OH· by steamed freeze-dried GE and similar results for dried and sun-dried GE (Table 5). The order of DPPH scavenging ability is as follows: SFD > SAD > SD > FFD > FD > FAD > SF9 with IC_50_ values of 22.68, 23.61, 38.07, 40.42, 47.52, 62.39 and 73.71 mg/mL. The DPPH scavenging abilities of fresh and directly dried GE were weaker than those of steam-dried GE. The SFD-, FFD- and FD-treated GE showed good ABTS scavenging ability, with IC_50_ values of 1.74, 2.00 and 2.06 mg/mL, while those of SF9, SD, FAD and SAD were less effective, with the half maximal inhibitory concentration (IC_50)_ values of 2.78, 3.05, 3.15 and 3.38 mg/mL. The scavenging ability of OH· was superior for dried-after-steaming GE, with IC_50_ values of 40.34, 42.71, 66.23 and 69.86 mg/mL in SD, SAD, SFD and SF9, which indicated that steaming can promote the enrichment of active ingredients (Table 3).

#### 2.2.2. α-Glucosidase Inhibition Assay

The results of analyzing the α-glycosidase inhibitory activities of extracts and the main components of GE showed that the GE extract of DFW-6 had the strongest α-glycosidase inhibitory ability, followed by DFW-3, with IC_50_ values of 0.13 and 0.29 mg/mL, and the worst inhibition activity belonged to the extract of DFW-2, with an IC_50_ value of 1.92 mg/mL (Table 5). Meanwhile, the high-potential inhibitory ability of the monomeric compounds to inhibit α-glycosidases was held by parishin C, HBA and parishin B, with IC_50_ values of 0.13, 0.16 and 0.19 mg/mL, respectively (Table 6).

### 2.3. Network Pharmacology and Molecular Docking

#### 2.3.1. Collection of Six Marker Components of GE- and T2DM-Related Targets

All targets of GAS (102), HBA (109), parishin E (102), parishin B (104), parishin C (101) and parishin A (101) were collected from the SwissTargetPrediction database. The targets of the compounds were pooled and integrated and 320 GE marker targets were collected after removing duplicates. A total of 15,336 T2DM-related disease targets were obtained from the GeneCards and Omim databases after de-duplication. Finally, 285 potential common targets of GE marker components and T2DM were obtained (Figure 2).

#### 2.3.2. PPI Analysis of Core Targets

The obtained 285 potential common targets of marker components with T2DM were transferred to the STRING platform (https://string-db.org/ accessed on 9 March 2024) for PPI analysis and visualized in CytoScape 3.9.1 software. Finally, an interaction network of GE marker components with T2DM common targets was constructed. The size and color of the nodes were based on degree values, while the thickness and color of the edges were based on the size of the combined score. There were 53 nodes (representing 53 potential common targets) and 651 edges (representing interactions between 651 potential common targets), as shown in Figure 3.

#### 2.3.3. Important and Relevant Target Networks

In total, twelve targets were screened based on the screening criteria for the degree value ≥ 718.5557784, median centrality ≥ 0.001858736 and proximity centrality ≥ 51, which indicated that these twelve targets were very important in this network and might be the key targets of GE marker components with T2DM. The gene names, degree values, median centrality (BC) and proximity centrality (CC) of the key targets are shown in Table 7. The top four targets ranked by degree value were GAPDH (degree = 148), TNF (degree = 142), AKT1 (degree = 140) and SRC (degree = 122).

#### 2.3.4. GO and KEGG Enrichment Analysis of Core Targets

The GO enrichment analysis of 53 intersection targets of the main components of GE obtained from the Metascape database was performed to investigate the potential function of GE for diabetes by transferring the 53 important targets, which were mainly involved in three areas (www.bioinformatics.com.cn/ accessed on 9 March 2024): biological processes (BP), cellular components (CC) and molecular functions (MF). The results revealed that 3840 BP, 338 CC and 587 MF existed. Furthermore, the results of the analysis for the top 10 degree values in each category were selected, plotted and represented in a histogram (Figure 4a). KEGG pathway enrichment analysis was performed by transferring 53 key targets to the Metascape database, which revealed 249 signaling pathways. The KEGG pathways with the top 20 degree values were selected for labeling and represented as bubble plots (Figure 4b).

#### 2.3.5. The “Component–Target–Disease (CTD)” Network

The network of the six components of GE, 53 important targets and the top 10 signaling pathways with T2DM was constructed using Cytoscape 3.9.1 software. However, the results revealed that there were no known direct connections between DPP4, ADA, IDO1, MPO, KDR, CYP19A1, AKR1B1, TYMS, EPHX2 and the top 10 signaling pathways (Figure 5). Interestingly, the CTD network of ATK1, PRKCA, CASP3, MAPK1, PTGS1, MMP9, PTGS2 and TNF showed more relevance, and all of these relevant targets have known studies on diabetes through a literature collection review, predicting that the six main components of GE may also treat T2DM by acting on these pathways.

#### 2.3.6. Molecular Docking Evaluation

The molecular docking methods were applied to validate the reliability of the interaction between the six main components of GE and the first six protein pathway target genes, and the docking with binding energies < −6 kcal/mol indicated that the complementary active ingredient and the target bind effectively in the natural state. The docking results showed that parishin A, parishin B, parishin C and parishin E could bind stably to multiple proteins at the same time, such as MMP9 (−6.34), CASP3 (−7.88), PRKCA (−7.31) and AKT1 (−6.10) (Table 8). The lower binding energy between the active ingredient and target protein receptors indicates the better affinity between them and the better structural stability (Table 9). The simulated docking results were visualized using PyMol 4.5.0 software, as shown in Figure 6.

## 3. Discussion

The Chinese government clarified that the total contents of GAS and HBA have been regarded as the hallmark components in GE for quality control in the 2020 edition of the Chinese Pharmacopoeia [18]. It is well known that different origins of GE in terms of genetics, plant origin, environmental factors, storage conditions, etc. will lead to differences in the content of its hallmark components [19]. Meanwhile, the current studies also confirmed that parishin analogs in GE play a crucial role in the treatment of central nervous system disease [20,21]. Therefore, the total amount of GAS, HBA and parishins should be considered comprehensively in the quality control study of GE. There is a conversion relationship between the GAS, parishins and HBA, which is a glycoside and metabolite of GAS and parishins [22,23]. In our study, we found that the content of HBA and parishin C in fresh GE was higher than that after steaming, and the content of GAS was extremely low; meanwhile, the contents of GAS, parishin E, parishin B and parishin A were increased in the steam-dried GE, which has been confirmed by the higher HBA content of 4.62 mg/g in DFW-1, the highest parishin C content of 5.35 mg/g in DFW-2 and the highest parishin A content of 10.89 mg/g in DFW-3. After steaming, the six components’ content of GE was higher than that of directly dried GE samples, and DFW-6 had the highest total content of 33.87 mg/g, while GAS, parishin E, parishin B and parishin A had the highest contents of 4.82 mg/g, 7.93 mg/g, 6.86 mg/g and 10.89 mg/g along with decreased HBA and parishin C contents, which is consistent with Wu et al.’s reported results [24]. It was pointed out that the vaporization process can inhibit the activity of β-D-glucosidase and the enzymatic deglycosylation of GAS, and then the asparagus-binding glycosides will be obtained by breaking the ester bond or benzyl ether bond to obtain GAS and HBA [25]. The exoenzyme digestion experiment also proved that carboxylesterase (CarE) and β-glucosidase (β-GC) can produce the de-glucose degradation of parishins in GE, and the results of the measurement of the activity of the two hydrolases, CarE and β-GC, confirmed that with the prolongation of the time of steaming, the activity of CarE and β-GC was gradually reduced, and the inactivation of the two hydrolases was achieved when the GE was steamed, which could protect the parishin and could also be used as a protective agent of GAS. It can play a role in the protection of balisenosides and GAS [26]. It also illustrates the feasibility and rationality of the inactivation and preservation of glycosides using the steaming GE process in ancient times.

Due to the different properties and sensitivities of the samples to antioxidant species, a single antioxidant assay cannot fully evaluate all the active substances in GE extracts. Thus, multiple antioxidant methods were conducted to validate the antioxidant capacity of GE processed with seven different drying method using the DPPH, ABTS and OH· assays [27]. Interestingly, the steam-dried GE showed better antioxidant activities than fresh directly dried GE, with the strongest antioxidant activity in the steam-freeze-dried GE sample, as shown in Table 6. A previous study has confirmed that the steamed GE displays better hypnotic and anxiolytic activities in mice compared to the fresh GE sample, which is consistent with our results [15]. Therefore, GE extract may indirectly act as an antioxidant defense by activating a series of biochemical pathways in the cellular antioxidant system, except for the direct scavenging mechanism of free radicals [28].

The treatment of T2DM patients for glycemic control usually involves α-glucosidase, which is a promising treatment for diabetes that regulates the activity of key carbohydrate-metabolizing enzymes to stabilize blood glucose levels [29]. α-glucosidase inhibitors are a new class of anti-diabetic drugs with a sugar structure, which have been widely used in clinics, including acarbose, voglibose and miglitol, This kind of drug can significantly reduce the postprandial blood glucose level in T2DM patients and reduce the occurrence of diabetes complications [30]. In the present study, the α-glucosidase enzyme inhibition and antioxidant activity of GE samples revealed a similar trend, namely, the steam-dried GE showed better α-glucosidase inhibition activities than fresh directly dried GE. Notably, all seven GE samples were able to inhibit the enzyme activity, and the inhibitory ability of its parishins’ compounds could also inhibit the α-Glucosidase enzyme activity in a better way than GAS and HBA (Table 5 and Table 6). Meanwhile, the GE lyophilized after steaming showed better glucose-lowering ability than the fresh samples, which may depend on the significant differences in the content changes of parishin C, HBA, parishin B and parishin A.

Recent studies have confirmed that the main pharmacodynamic components of GE are GAS, HBA and parishins compounds, which are present in high proportion levels [31]. Meanwhile, studies have confirmed that GAS [32] and parishins compounds possess significant therapeutic effects on diabetes and show a good inhibitory ability regarding the α-glucosidase enzyme, indicating that the six main components in GE can be potential candidates for the treatment of T2DM. In this study, 238 potential target genes were obtained for the treatment of T2DM through screening six main compounds of GE using a network pharmacology approach. Among them, ATK1, PRKCA, TNF, CASP3, MMP9, PTGS2 and MAPK1 were potential key genes associated with the pathways for the treatment of T2MD. AKT1, protein kinase Bα, is a serine/threonine kinase that plays a key role in intracellular signaling and is a core component of the PI3K/Akt signaling pathway involved in diabetic disease. The PI3K/Akt signaling pathway is involved in a variety of physiological processes and is an important signaling pathway for the development of many diseases, especially tumors, diabetic disease and cardiovascular disease. It can regulate the glucose transport, the β-cell secretion of hormones and the transcription of insulin genes [33]. Moreover, the activation of Akt can promotes the glucose transporter protein 2 (GLUT 2) transport, it regulates glucose metabolism and it is an important pathway in the liver that allows for insulin regulation [34]. In the meantime, the PI3K/Akt signaling pathway regulates the PRKCA and TNF-α proteins, which can reduce the production of inflammatory factors and decrease pancreatic islet β-cell apoptosis through the downregulation of the expression of pancreatic islet β-cells and the regulation of the disorders of lipid metabolism and then achieve the treatment purpose of T2DM [35]. The AMPK signaling pathway is an important metabolic pathway that regulates various cellular processes. Studies have shown that the activation of the AMPK pathway and downregulatoon of the MMP9 protein can be therapeutic for sciatic nerve disease in diabetic mice [36]. When the body is continuously exposed to inflammatory responses, the BL001/LRH-1/NR5A2 axis aspect protects pancreatic islet cells from cytokines. The LRH-1/PTGS2/PGE2/PTGER1 signaling axis is a key pathway mediating the survival properties of BL001, and the expression of PTGS2 is a key target for the activation of this pathway, which may be conveyed in part through immune cell signaling between pancreatic islet cells to convey the reversal of type 1 diabetes symptoms [37]. The oxidative metabolism-dependent stabilization and activation of MAPK1 and MAPK3 accelerates gastric emptying in diabetic patients, and the inhibition of metabolism of this pathway may alleviate hyperglycemic symptoms in them [38]. In addition, molecular docking simulations further screened that GAS, parishin A, parishin B, parishin C and parishin E exhibited a strong binding ability to the corresponding targets, which indicated that the linkage between the selected core active ingredients and the core diabetes-related target genes was stable, suggesting that the marker active compounds in GE could bind to the key diabetes-related genes and induce antidiabetic activity.

## 4. Materials and Methods

### 4.1. Chemicals

P-nitrophenyl-α-D-glucopyranoside (PNPG) (purity ≥ 99%) was obtained from Shanghai McLean Biochemical Technology Co. (Shanghai, China). Gastrodin (purity > 99.02%), p-Hydroxybenzyl alcohol (purity > 99.02%), Parishin A (purity > 99.02%), Parishin B (purity > 99.02%), Parishin C (purity > 99.02%) and Parishin E (purity > 99.02%) standards were purchased from Chengdu Pusi Biotechnology Co. (Chengdu, China); 2,2-diphenyl-1-picrylhydrazyl radical (DPPH), sodium carbonate, diammonium salt (ABTS), α-glucosidase, 3,5-dinitro salicylic acid and potassium sodium tartrate were purchased from Beijing Solarbio Technology Co. (Beijing, China).

### 4.2. Analysis of the Content of Marker Components of GE

#### 4.2.1. Sample Collection and Handling

The fresh GE tubers were purchased from two characteristic production areas, Dafang and Dejiang, Guizhou Province, China. A total of four sets of GE samples, Dafang wild-imitated cultivation *G. elata Bl. F. elata* (DFW), Dafang bag-planted *G. elata Bl. F. elata* (DFB), Dejiang wild-imitated cultivation *G. elata Bl. F. Glauca S Chow* (DJW) and Dejiang bag-planted *G. elata Bl. F. Glauca S Chow* (DJB), were collected. The fresh GE was directly cut into thin slices of about 3 mm. The GE samples of each group were subjected to different drying methods, as shown in the Table 4. Drying: placed in the oven for 28 h at 50 °C; Air-drying: put in a cool and dry room for 50 h at 10 °C; Freeze-drying: placed in the −80 °C freeze-dryer for 48 h; Steaming for nine times: steam 15 min, put into the oven for 24 h at 50 °C and repeat nine times. Four groups of 28 asparagus samples with different drying treatments were obtained.

#### 4.2.2. Preparation of Sample and Standard Solutions

The dried products were pulverized using a high-speed Chinese medicine pulverizer and then passed through a 60-mesh sieve. A precisely weighed sample (2.0 g) was immersed in a 50 mL volumetric flask and 60% methanol was added to fix the volume; ultrasonic extraction for 60 min and then 60% methanol were used to make up for the loss of weight. All solutions were filtered through 0.22 µm filter membrane, and then precisely 10 µL was injected into the HPLC system [13]. The GAS (20.16 mg), HBA (20.56 mg), parishin E (28.08 mg), parishin B (20.60 mg), parishin C (20.48 mg) and parishin A (27.40 mg) standards were dissolved in 60% methanol, and the standard solutions were obtained in a 5 mL volumetric bottle at a constant capacity. All standard solutions were stored in a −4 °C refrigerator for backup.

#### 4.2.3. HPLC System Conditions

The HPLC analysis of GE samples was performed on an Agilent 1260 series system (Agilent Technologies, Santa Clara, CA, USA) coupled with a phenomenex Luna C18 column (250 × 4.6 mm, 5 μm) by using the gradient elution method, as follows: 0–10 min, 97–90% B; 10–18 min, 90–88% B; 15–25 min, 88–82% B; 25–40 min, 82% B; and 40–42 min, 82–5% B (A, ddH_2_O + 0.1% formic acid; B, acetonitrile). The flow rate was set at 1.0 mL/min, and the detection wavelength was 220 nm. The column temperature was 30 °C and the injection volume was 10 µL [18]. The precision, repeatability, stability and accuracy of HPLC methods were verified using the GE sample, as described in Appendix A.

#### 4.2.4. Investigation of the Linear Relationship of the Standard Solution

The stock standard solutions with the concentrations of 4.032 mg/mL of GAS, 4.112 mg/mL of HBA, 5.616 mg/mL of parishin E, 4.12 mg/mL of parishin B, 4.096 mg/mL of parishin C and 5.48 mg/mL of parishin A were diluted with the following method: precisely absorb 0.1 mL of the standard reserved solution to 1, 5, 10, 20, 50 and 100 mL brown measuring bottles and dilute the volume to the scale with 60% methanol. The standard solution in a series of concentrations was injected into HPLC with the chromatographic conditions in Section 4.2.3. Linearity is determined as a function of the peak area (y) and corresponding concentration (x, mg/mL). The standards were quantified by the linear equations obtained. The LOD was considered to be the concentration that produced a signal-to-noise (S/N) ratio of 3, and the LOQ was defined at an S/N ratio of 10.

### 4.3. Antioxidant Activities and α-Glucosidase Inhibition Assay

#### 4.3.1. Sample Preparation

The DFW sample (2 g) was extracted with 10 mL of 60% methanol for 1 h by ultrasonic extraction, and then the supernatant was centrifuged and diluted into 200 mg/mL, 100 mg/mL, 50 mg/mL, 25 mg/mL, 12.5 mg/mL and 6.25 mg/mL sample solutions for the bioactivity study.

#### 4.3.2. DPPH Scavenging Test

The DPPH scavenging activity of GE samples was determined according to the manufacturer’s instruction. In total, 100 µL of the sample solution (200 mg/mL) and 100 µL of the DPPH solution were sequentially added to 96-well plate and reacted for 30 min at 25 °C under dark ambient conditions, and the absorbance was measured at 517 nm. The ultrapure water and methanol were used to replace the GE sample and DPPH solution in the blank (Ac) and sample blank groups, respectively. The inhibition rate was calculated using the following formula [39]: (Ac − As)/Ac × 100%.

#### 4.3.3. ABTS· Scavenging Test

The ABTS scavenging activity test was performed according to the reported literature [7]. The method was slightly modified by taking 10 µL of the sample solution at a concentration of 12.5 mg/mL and adding 170 µL of ABTS radical solution into a 96-well plate, which was reacted for 6 min with light closed at room temperature. Then, the absorbance was measured at 405 nm.

#### 4.3.4. OH· Scavenging Test

The OH· scavenging test was performed according to Li’s reported method [40]. The sample solution (200 mg/mL, 150 µL), 650 µL of the OH· working solution and 450 µL of purified water were mixed in a centrifuge tube, and the mixture was reacted in a 37 °C constant temperature oven for 60 min. After the reaction, the mixture was centrifuged, and the absorbance of the supernatant was measured at 536 nm. The working solution and pure water were used as the blank control for all the blank groups.

#### 4.3.5. α-Glucosidase Inhibition Assay

Appropriate improvements were made with reference to the α-glucosidase assay [41]. The extract of GE at a concentration of 10 mg/mL was prepared and added to the 96-well plate (25 µL), followed by the reaction of adding 50 µL of α-glucosidase solution (0.2 U/mL, pH 6.8) for 10 min. Then, 25 µL of PNPG solution (5 mg/mL) was added for 10 min under light protection, and Na_2_CO_3_ termination solution was used to stop the reaction. The absorbance of the samples was then measured at 405 nm. The inhibitory ability of Acarbose was used as a reference.

### 4.4. Network Pharmacology Analysis of Potential Mechanisms of GE Components for Diabetes Treatment

#### 4.4.1. Databases and Software

The following databases and software were used to build the network pharmacology and molecular docking: the SwissTargetPrediction database (http://www.SwissTargetPrediction.ch/ accessed on 9 March 2024), the Human GeneCards (GeneCards, https://www.genecards.org/ accessed on 9 March 2024), the OMIM database (http://www.omim.org/ accessed on 9 March 2024), the Metascape Database (https://jvenn.troulouse.inra.fr/ accessed on 9 March 2024), Cytoscape version 3.9.1 and the Venn Online Interactive Tool (http://www.example.com/ accessed on 9 March 2024).

#### 4.4.2. Common Target Prediction of GE Components with T2DM

The marker component-related disease targets were obtained from the TCM, Symmap, CTD and SwissTargetPrediction databases [42]. T2DM-related targets were obtained from the GeneCards and OMIM databases [43]. And then, the two target sets were compared for potential common targets of GE marker components with T2DM. Detailed target information is provided in Appendix A.

#### 4.4.3. Construction of the Protein–Protein Interaction Network and Screening

The protein–protein interaction (PPI) analysis was performed by importing the predicted potential common targets into the STRING database (https://www.tring-db.org/ accessed on 11 March 2024) with a confidence level of 0.7 for the better authenticity and accuracy of data [44]. Then, the analysis results were imported into CytoScape 3.9.1 software, and the degree and combined score values were designed as the size of the number of nodes and the thickness of the edges, respectively. Finally, a network diagram of common protein interactions was obtained [45]. The Gene names, degree value, betweenness centrality (BC) and closeness centrality (CC) of targets are shown in Appendix A.

#### 4.4.4. GO Function and KEGG Pathway Enrichment Analysis

The Metascape database (https://www.metascape.org/ accessed on 11 March 2024) was conducted to analyze the GO function and KEGG pathway enrichment of key targets. The species selection was human, the minimum overlap was set to 3, the *p*-value cutoff was set to *p* < 0.05, and the minimum enrichment value was set to 1.5 and then the bubble map of the GO function and KEGG pathway was obtained. A histogram analysis was performed for the top 10 GO functions after sorting the gene proportions from largest to smallest [46].

#### 4.4.5. Construction of “Drug–Target–Pathway–Disease” Network

The “drug–target–pathway–disease” network was mapped using Cytoscape 3.9.1 software. The targets of the KEGG pathway were obtained and analyzed, and the top 10 pathway-related targets with the smallest *p*-value were included in the analysis, which could better clarify the association between the key targets and the pathways, and the targets that were not related to these 10 pathways were removed.

#### 4.4.6. Molecular Docking of Marker Components of GE

To study the combining capacity between the key targets and GE marker components, the AutoDock software (https://vina.scripps.edu/ accessed on 20 May 2024) was used to perform molecular docking [47]. The 3D structures of the marker components of GE were obtained from the PubChem database, and the 3D structures of core proteins were obtained from the Protein Data Bank (https://www.rcsb.org/ accessed on 20 May 2024). The AutoDock and Pymol software were used to pre-process the target protein, and then the AutoDock 1.2.0 software was used for batch molecular docking. Finally, the docking results were imported into for analysis and visualization [48]. The crystal structures of the target proteins were downloaded from the PDB database [49]. Combined with the review of literature related to the targets of asparagus and diabetes, the first six protein pathways that were highly relevant and the six marker components of GE were screened for molecular docking. The binding affinity (kcal/mol) value represents the binding free energy of the component regarding the target protein; the higher the absolute value of the binding free energy, the more stable the binding of the ligand to the receptor.

### 4.5. Statistical Analysis

HPLC measurements and activity tests of GE samples were performed in triplicate. Mean values were calculated from the data of three independent experiments, and IC_50_ values were calculated using the software IBM SPSS Statistics 26.0. The results correspond to the estimated mean ± standard deviation of the extract.

## 5. Conclusions

The processing process of medicinal materials has a great influence on its marker components. This study confirmed that the steam-frozen and steam-oven-dried methods for fresh GE can be used in the processing of GE products based on the higher marker component contents and the potential antioxidant and α-glycosidase inhibitory ability. Moreover, the marker components of GE are highly relevant to the target’s proteins of T2DM treatment through network pharmacology analysis. Notably, the marker components of GE also showed a strong ability to bind to the target proteins of the T2DM-related potential pathway through molecular docking methods. Overall, the steam-lyophilized GE samples had a higher marker components content along with an excellent antioxidant capacity and α-glycosidase inhibitory activity. The parishins’ constituents with strong glycosidase inhibitory activity have the potential to be exploited for diabetes treatment and the possible therapeutic mechanisms of their pharmacodynamic constituents that are different from each other, suggesting it is worthwhile to investigate the synergistic therapeutic effects on the treatment of T2DM.

## Figures and Tables

**Figure 1 ijms-25-10498-f001:**
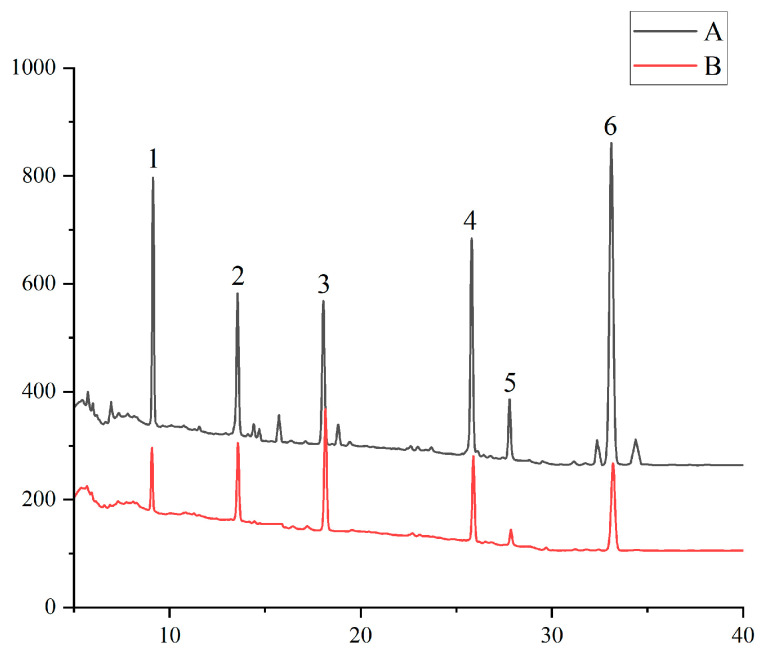
The HPLC fingerprints of the GE (DFW-5) (A) and the mixed standards (B). Peaks and concentrations corresponding to mixed standards: 1, GAS (0.2016 mg/mL); 2, HBA (0.2056 mg/mL); 3, parishin E (0.2808 mg/mL); 4, parishin B (0.2060 mg/mL); 5, parishin C (0.2048 mg/mL); 6, parishin A (0.1370 mg/mL).

**Figure 2 ijms-25-10498-f002:**
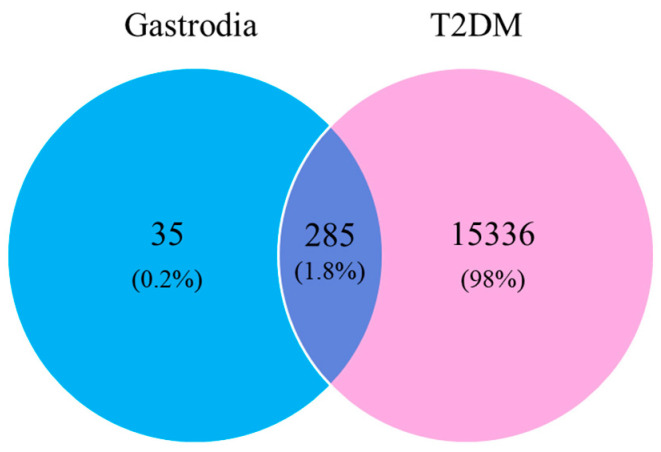
Venn diagram of related targets of six maker components of GE and T2DM.

**Figure 3 ijms-25-10498-f003:**
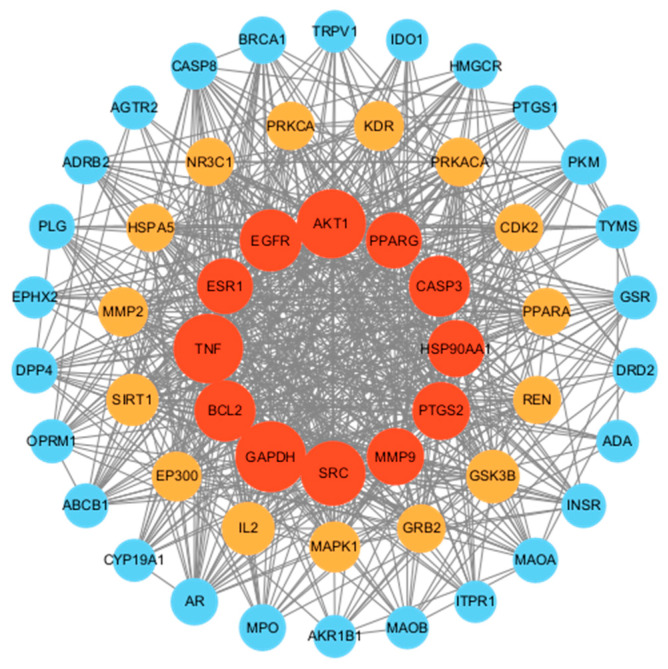
PPI network and core target genes.

**Figure 4 ijms-25-10498-f004:**
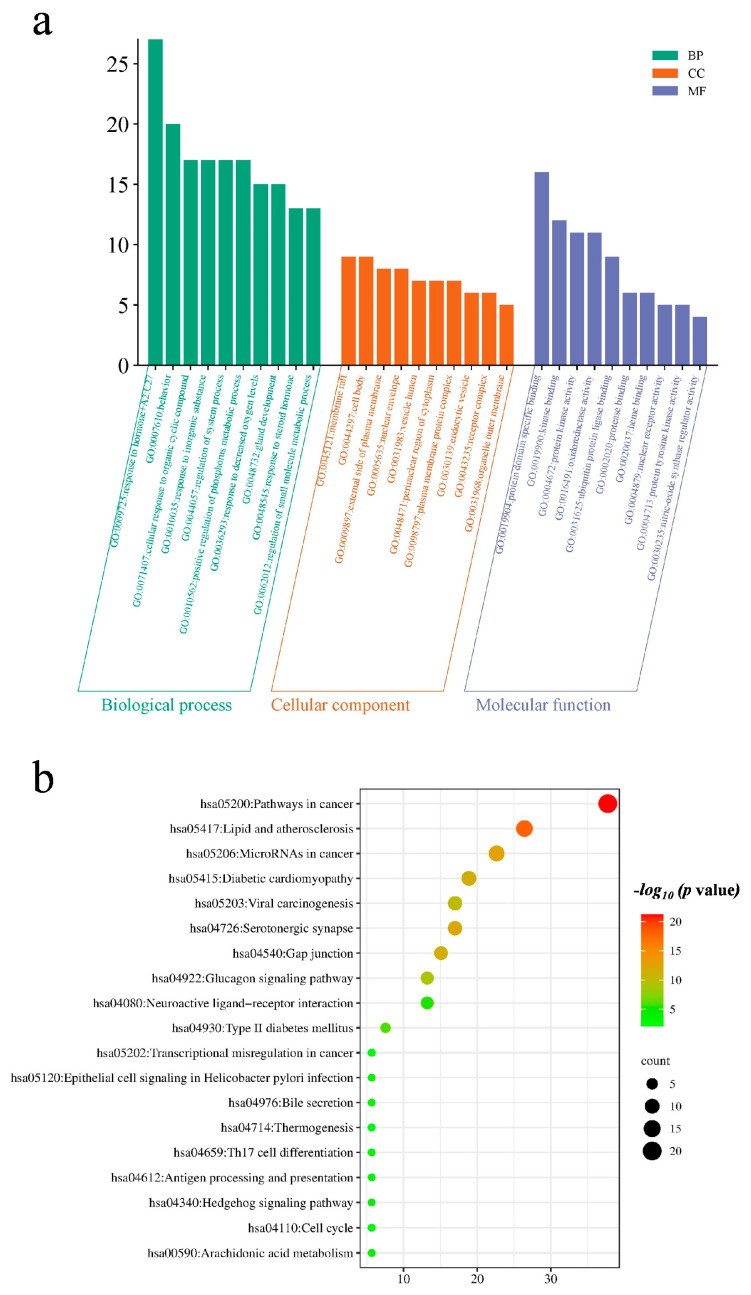
Enrichment analysis of core targets: GO function (**a**) and KEGG pathway (**b**).

**Figure 5 ijms-25-10498-f005:**
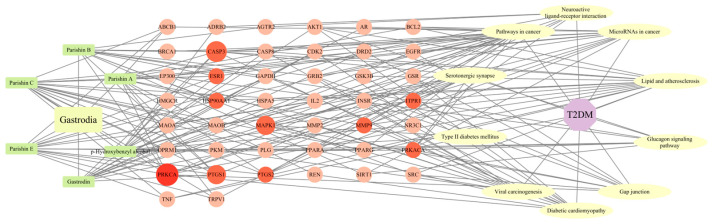
The “Component–Target–Disease” network.

**Figure 6 ijms-25-10498-f006:**
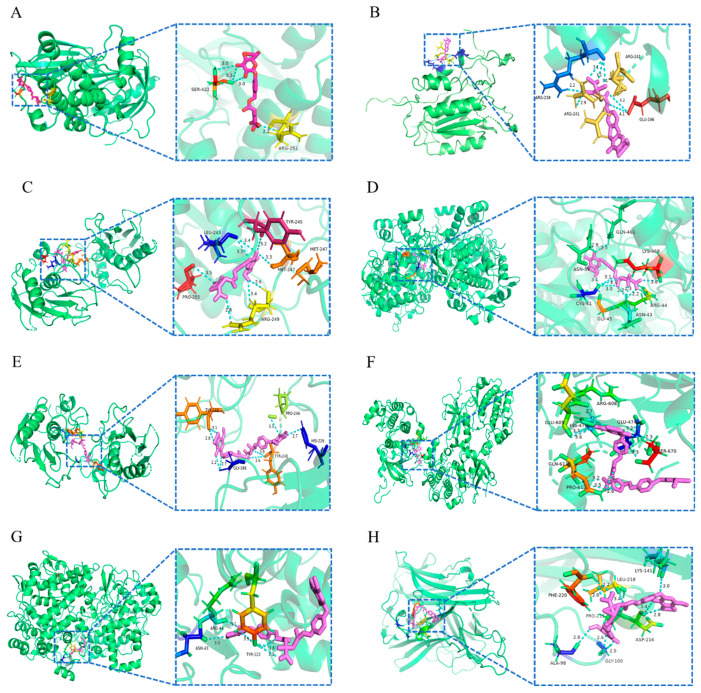
Molecular docking results of the lowest binding energy in each target with the selected bioactive compound (violet) of GE. (**A**) Parishin E, AKT1, −6.1 kcal/mol; (**B**) Parishin E, CASP3, −6.26 kcal/mol; (**C**) Parishin E, MMP9, −8.95 kcal/mol; (**D**) Parishin E, PTGS2, −7.69 kcal/mol; (**E**) Parishin B, AKT1, −7.21 kcal/mol; (**F**) Parishin B, CASP3, −7.88 kcal/mol; (**G**) Parishin B, PRKCA, −6.08 kcal/mol; (**H**) Parishin C, MMP9, −9.4 kcal/mol; (**I**) Parishin C, PRKCA, −7.31 kcal/mol; (**J**) Parishin C, PTGS2, −7.1 kcal/mol; (**K**) Parishin C, TNF, −6.33 kcal/mol; (**L**) Parishin A, MMP9, −6.34 kcal/mol.

**Table 1 ijms-25-10498-t001:** The marker components in GE.

Active Ingredients	CAS	Chemical Structure
GAS	62499-27-8	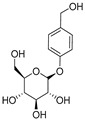
HBA	623-05-2	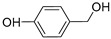
Parishin E	952068-57-4	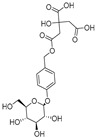
Parishin B	174972-79-3	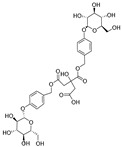
Parishin C	174972-80-6	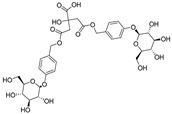
Parishin A	62499-28-9	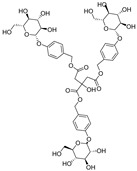

**Table 2 ijms-25-10498-t002:** HPLC analyzed the linear regression data of the six marker components.

Analytes	Regression Equations	Linear Ranges (mg/mL)	*R* ^2^	LOD (mg/mL)	LOQ (mg/mL)
GAS	y = 16,369x + 46.2	0.0040–0.40320	0.9996	0.00039815	0.00132716
HBA	y = 32,748x + 7.8054	0.0041–0.4112	0.9994	0.00007025	0.00023417
Parishin E	y = 10,822x + 13.016	0.0056–0.5616	0.9999	0.00018952	0.00063172
Parishin B	y = 13,174x + 8.4432	0.0041–0.4120	0.9996	0.00017482	0.00058274
Parishin C	y = 13,946x + 8.627	0.0041–0.4096	0.9993	0.00017356	0.00057853
Parishin A	y = 18,981x + 50.267	0.0055–0.5480	0.9996	0.00154903	0.00516344

LOD: limit of detection; LOQ: limits of quantification.

**Table 3 ijms-25-10498-t003:** The contents of marker components of GE in different drying methods.

No.	Mean Content in Samples (*n* = 3) (mg/g)						
	GAS	HBA	Parishin E	Parishin B	Parishin C	Parishin A	Total Content
DFW-1	1.69 ± 0.04	4.62 ± 0.10	4.46 ± 0.01	2.69 ± 0.13	3.04 ± 0.15	4.85 ± 0.13	21.35 ± 0.13
DFW-2	0.42 ± 0.01	4.61 ± 0.15	3.55 ± 0.07	1.15 ± 0.02	5.35 ± 0.12	2.05 ± 0.13	17.13 ± 0.29
DFW-3	0.73 ± 0.03	3.27 ± 0.07	4.44 ± 0.04	1.83 ± 0.06	2.31 ± 0.15	8.31 ± 0.25	20.88 ± 0.34
DFW-4	3.60 ± 0.08	1.54 ± 0.07	6.78 ± 0.06	5.13 ± 0.08	1.00 ± 0.04	7.59 ± 0.22	25.64 ± 0.22
DFW-5	2.93 ± 0.04	1.59 ± 0.03	6.50 ± 0.05	5.12 ± 0.05	0.98 ± 0.03	6.84 ± 0.17	23.96 ± 0.06
DFW-6	4.82 ± 0.14	1.98 ± 0.03	7.93 ± 0.14	6.86 ± 0.10	1.39 ± 0.05	10.89 ± 0.94	33.87 ± 0.70
DFW-7	2.07 ± 0.06	0.90 ± 0.03	4.78 ± 0.13	5.02 ± 0.24	1.05 ± 0.08	9.76 ± 0.22	23.58 ± 0.15
DFB-1	0.24 ± 0.02	2.68 ± 0.16	2.04 ± 0.05	1.43 ± 0.08	0.88 ± 0.02	4.74 ± 0.08	11.98 ± 0.36
DFB-2	0.28 ± 0.04	2.43 ± 0.10	1.80 ± 0.06	0.59 ± 0.01	0.57 ± 0.02	2.02 ± 0.01	7.69 ± 0.04
DFB-3	0.04 ± 0.01	1.57 ± 0.09	1.43 ± 0.02	0.88 ± 0.02	0.33 ± 0.00	6.80 ± 0.17	11.05 ± 0.06
DFB-4	1.70 ± 0.06	1.29 ± 0.06	6.20 ± 0.13	3.37 ± 0.09	0.57 ± 0.01	3.68 ± 0.11	16.81 ± 0.16
DFB-5	1.04 ± 0.05	1.01 ± 0.01	4.78 ± 0.07	2.31 ± 0.11	0.31 ± 0.01	2.20 ± 0.03	11.65 ± 0.27
DFB-6	1.86 ± 0.06	1.40 ± 0.02	8.37 ± 0.21	3.82 ± 0.08	0.67 ± 0.01	4.18 ± 0.05	20.30 ± 0.25
DFB-7	2.94 ± 0.07	0.63 ± 0.01	7.20 ± 0.20	3.89 ± 0.11	0.98 ± 0.02	3.89 ± 0.07	19.53 ± 0.52
DJW-1	0.21 ± 0.01	2.67 ± 0.13	1.76 ± 0.12	2.00 ± 0.05	2.09 ± 0.06	4.00 ± 0.10	12.73 ± 0.11
DJW-2	0.01 ± 0.00	2.53 ± 0.07	0.90 ± 0.04	0.33 ± 0.00	3.50 ± 0.12	0.95 ± 0.01	8.22 ± 0.01
DJW-3	0.02 ± 0.01	1.87 ± 0.04	1.99 ± 0.11	1.80 ± 0.00	0.25 ± 0.02	6.82 ± 0.12	12.75 ± 0.09
DJW-4	2.42 ± 0.09	0.53 ± 0.03	5.40 ± 0.04	6.75 ± 0.08	1.12 ± 0.02	11.35 ± 0.22	27.57 ± 0.24
DJW-5	1.95 ± 0.03	0.56 ± 0.03	3.80 ± 0.06	5.57 ± 0.17	0.91 ± 0.01	8.68 ± 0.26	21.47 ± 0.39
DJW-6	2.88 ± 0.08	0.58 ± 0.03	4.53 ± 0.09	6.30 ± 0.07	1.24 ± 0.01	13.57 ± 0.23	29.10 ± 0.23
DJW-7	3.22 ± 0.17	0.25 ± 0.04	4.55 ± 0.10	8.07 ± 0.13	1.61 ± 0.02	11.11 ± 0.16	28.81 ± 0.57
DJB-1	0.14 ± 0.02	1.39 ± 0.04	2.14 ± 0.02	1.62 ± 0.04	4.65 ± 0.10	1.25 ± 0.00	11.19 ± 0.03
DJB-2	0.01 ± 0.00	2.21 ± 0.03	1.01 ± 0.01	0.81 ± 0.00	4.32 ± 0.06	1.84 ± 0.05	10.20 ± 0.05
DJB-3	0.19 ± 0.00	2.38 ± 0.02	1.35 ± 0.00	0.78 ± 0.01	2.10 ± 0.06	2.42 ± 0.09	9.22 ± 0.09
DJB-4	1.54 ± 0.02	0.84 ± 0.01	3.66 ± 0.07	4.67 ± 0.16	0.81 ± 0.02	7.04 ± 0.26	18.54 ± 0.16
DJB-5	1.13 ± 0.04	0.64 ± 0.00	2.58 ± 0.06	3.16 ± 0.10	0.54 ± 0.00	5.11 ± 0.22	13.16 ± 0.07
DJB-6	1.94 ± 0.04	0.64 ± 0.01	4.23 ± 0.10	5.41 ± 0.18	0.95 ± 0.00	7.63 ± 0.14	20.80 ± 0.14
DJB-7	2.84 ± 0.03	0.36 ± 0.000	3.82 ± 0.11	4.44 ± 0.10	1.03 ± 0.00	5.20 ± 0.16	17.69 ± 0.49

Dafang wild-imitated cultivation *G. elata Bl. F. elata* (DFW), Dafang bag-planted *G. elata Bl. F. elata* (DFB), Dejiang wild- imitated cultivation *G. elata Bl. F. Glauca S Chow* (DJW) and Dejiang bag-planted *G. elata Bl. F. Glauca S Chow* (DJB). The results correspond to the estimated mean ± standard deviation of the extract.

**Table 4 ijms-25-10498-t004:** Drying method of GE samples.

NO.	Drying Method	Sample ID
1	fresh drying (FD)	DFW-1, DFB-1, DJW-1, DJB-1
2	fresh air-drying (FAD)	DFW-2, DFB-2, DJW-2, DJB-2
3	fresh freeze-drying (FFD)	DFW-3, DFB-3, DJW-3, DJB-3
4	steamed drying (SD)	DFW-4, DFB-4, DJW-4, DJB-4
5	steamed air-drying (SAD)	DFW-5, DFB-5, DJW-5, DJB-5
6	steamed freeze-drying (SFD)	DFW-6, DFB-6, DJW-6, DJB-6
7	steamed and dried nine times (SF9)	DFW-7, DFB-7, DJW-7, DJB-7

Drying: placed in the oven for 28 h at 50 °C; Air-drying: put samples in a cool and dry room for 50 h at 10 °C; Freeze-drying: placed in the −80 °C freeze-dryer for 48 h; Steaming for nine times: steam 15 min, put into the oven for 24 h at 50 °C and repeated nine times.

**Table 5 ijms-25-10498-t005:** IC_50_ of antioxidant activities and α-glucosidase inhibition assay.

Drying Type	Sample ID	IC_50_ (mg/mL)			
		DPPH	ABTS	OH·	α-Glucosidase
FD	DFW-1	47.52 ± 0.45	2.06 ± 0.10	131.92 ± 15.64	0.49 ± 0.05
FAD	DFW-2	62.39 ± 2.55	3.15 ± 0.08	122.65 ± 13.73	1.93 ± 0.08
FFD	DFW-3	40.42 ± 2.89	2.00 ± 0.11	116.04 ± 9.87	0.29 ± 0.03
SD	DFW-4	38.07 ± 1.80	3.05 ± 0.25	42.71 ± 3.79	0.58 ± 0.05
SAD	DFW-5	23.61 ± 1.55	3.38 ± 0.14	69.86 ± 6.42	1.47 ± 0.05
SFD	DFW-6	22.68 ± 1.88	1.74 ± 0.07	40.34 ± 4.37	0.13 ± 0.01
SF9	DFW-7	73.71 ± 4.30	2.78 ± 0.10	66.23 ± 5.64	0.34 ± 0.04

The results correspond to the estimated mean ± standard deviation of the extract.

**Table 6 ijms-25-10498-t006:** IC_50_ of α-glucosidase inhibition assay of GE components.

	GAS	HBA	Parishin E	Parishin B	Parishin C	Parishin A	Acarbose
IC50 (mg/mL)	1.23 ± 0.09	0.16 ± 0.03	0.54 ± 0.04	0.13 ± 0.03	0.19 ± 0.03	0.24 ± 0.05	0.02 ± 0.01

The results correspond to the estimated mean ± standard deviation of the extract.

**Table 7 ijms-25-10498-t007:** Gene names, degree value, betweenness centrality (BC) and closeness centrality (CC) of key targets.

Name	BC	CC	Degree Value
GAPDH	8172.014	0.002404	148
SRC	7812.498	0.002252	122
TNF	6068.216	0.002353	142
AKT1	5555.481	0.002347	140
EGFR	3012.723	0.002179	110
PTGS2	2312.372	0.002092	92
BCL2	2281.916	0.002155	106
CASP3	1658.903	0.002146	106
PPARG	1518.005	0.002033	84
MMP9	1400.578	0.002062	90
HSP90AA1	1337.491	0.002062	89
ESR1	1259.86	0.002049	83

**Table 8 ijms-25-10498-t008:** Molecular docking binding energies.

Compound	Protein and Binding Energy (kcal/mol)	
Gastrodin	AKT1	−4.35
	MMP9	−5.1
	TNF	−4.43
p-Hydroxybenzyl alcohol	MMP9	−5.3
	TNF	−4.63
Parishin E	AKT1	−6.1
	CASP3	−6.26
	PRKCA	−5.34
	MMP9	−8.95
	PTGS2	−7.69
	TNF	−4.97
Parishin B	AKT1	−7.21
	CASP3	−7.88
	PRKCA	−6.08
	MMP9	−4.4
	TNF	−5.33
Parishin C	AKT1	−5.03
	CASP3	−5.58
	MMP9	−9.4
	PRKCA	−7.31
	PTGS2	−7.1
	TNF	−6.33
Parishin A	AKT1	−4.55
	CASP3	−5.11
	MMP9	−6.34
	TNF	−4.23

**Table 9 ijms-25-10498-t009:** Table of the binding energy (kcal/mol) and interacting residues of key targets and active compounds.

Analytes	Gene Names	Vina Score	Interacting Residues
Parishin E	AKT1	−6.1	SER-422, ARG-251
	CASP3	−6.26	ARG-241, ARG-238, ARG-246
	MMP9	−8.95	PRO-225, ARG-249, MET-247, TYR-245, LEU-243
	PTGS2	−7.69	CYS-41, CLY-45, ASN-43, ARG-44, LYS-468, GLN-461, ASN-39
Parishin B	AKT1	−7.21	ARG-251, SER-246, ARG-249
	CASP3	−7.88	SER-267, ILE-265, ASN-240, PRO-263
	PRKCA	−6.08	VAL-664, GLU-474, TYR-419, LEU-364, HIS-665
Parishin C	MMP9	−9.4	TYR-248, GLY-186, TYR-248, PRO-246, HIS-226
	PRKCA	−7.31	PRO-612, SER-670, GLU-474, ARG-608, GLU-609, HIS-476, GLN-611
	PTGS2	−7.1	ARG-44, ASN-43, TYR-122
	TNF	−6.33	ALA-98, GLY-100, ASP-216, LYS-141, LEU-218, PHE-220, PRO-215
Parishin A	MMP9	−6.34	ASP-201, GLY-176, GLN-126, ARG-162, ASP-163, ARG-162

## Data Availability

The original contributions presented in the study are included in the article/Appendix A, further inquiries can be directed to the corresponding author.

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
