# Peer review of "Investigation of the Pharmacodynamic Components of Gastrodia elata Blume for Treatment of Type 2 Diabetes Mellitus through HPLC, Bioactivity, Network Pharmacology and Molecular Docking"

_ijms, 2024, doi:10.3390/ijms251910498_

Round 1

Reviewer 1 Report

Comments and Suggestions for Authors

Greetings to the authors, I have read your manuscript and my comments are as follows:
The introduction is interesting and detailed, however the authors do not mention the possible complications of diabetes mellitus, consequently they should add a paragraph in which they shortly remind the readers of potential complications such as cardiovascular complications, for example coronary artery disease (may refer to 10.3389/fendo.2018.00002), conduction disturbances (authors may refer to 10.3390/medicina57050441), Infectious complications (authors may refer to 10.4103/2230-8210.94253) and so on .

Authors should expand on the discussion regarding α-glucosidase  and α- amylase by mentioning their role and mechanism of action as well as how current pharmacological therapies effect these enzymes since they play a central role in the discussion of the manuscript.

I must say that the figures are interesting and look quite good, are they the authors own work ?

In the conclusion the authors make certain statements about insomnia and headaches from lines 444 to 446, this is not the central role of the manuscript and this part of the conclusions should be rephrased.

Comments on the Quality of English Language

Minor corrections are needed overall.

Reviewer 2 Report

Comments and Suggestions for Authors

- On the introduction section, rosiglitazone belongs to the thiazolidinediones pharmacological family which is further referred in the same sentence. It is also not correct to refer to oral antidiabetic drugs as “oral hypoglycemic agents” since not all these drugs directly act in decreasing glycemia values.

- Reference 9 is not adequate to support the first sentence of the introduction’s second paragraph. The study is focused on the effects of GE in depression, and it not extensively study or mention the effects on “convulsions, epilepsy, tetanus, vertigo and paralysis”. Please find more adequate references to support that statement and review all references along the manuscript.

- On introduction, paragraph 3, the last sentence that appears after reference 14 should not be there. It is out of context considering the information previously provided on that paragraph

- Please review and add proper references to the information provided on the last paragraph of the introduction sentence

- In figure 1, the concentrations of the standards that are injected into the HPLC system and to which the peaks shown correspond are those described in section 4.2.2? Please add this information to the figure caption.

- Do the concentrations of each analyte presented in figure 1 correspond to any concentration of the linear range presented on table 2? Please provide more information regarding the construction of the calibration curve, limits of quantification, etc… used to quantify each analyte by HPLC.

- On first sentence of section 2.1.2, please correct the mention to section 3.1.1. Please, also describe the meaning of the several presented abbreviations of that section. Also add the meaning of the abbreviations presented on each table on footnotes of each table.

- Shouldn't it be “market” instead of “marker”?  Otherwise, what do you mean by “marker”? That concept made me confused throughout all manuscript…

- The drying method provided on table 9 are not in accordance with the methods described in the text right before the table appears. Since the preparation of GE is such a crucial step to all the outcomes of the presented study, a more accurate description of it must be provided. Please revise.

- On section 4.2.2, did the authors directly inject a pure methanolic solution to the HPLC system? And how did the standard solutions were prepared? Did the injection of the solutions was performed right after preparation or there were no concerns regarding stability issues? Please clarify

- On section 4.2.3., please provide the composition of the mobile phase (A and B eluents).

- Though out all manuscript, to the results that are statistically analyzed and that are then presented as mean values, standard deviation or standard error of the mean values should be added for a better analysis of the results (also add that it to section 4.5)

- In the conclusion section, the first two sentences seem not fit in with the purpose of this section. Besides it, conclusions generally shouldn't contain references... Please revise all section

Comments on the Quality of English Language

- Overall, the english language must be improved (all sections, including abstract). Abstract must be particularly improved to make more sense.

Round 2

Reviewer 2 Report

Comments and Suggestions for Authors

Regarding the HPLC method validation, the authors did not provide the calibration range and respective calibration curve equation. Therefore, my  question in the first revision "Do the concentrations of each analyte presented in figure 1 correspond to any concentration of the linear range presented on table 2?" continue not to be answered. Additionally, the data regarding the anaysis of CV and bias of the quality control samples in the intraday and interday precision and acuracy evaluation is also not provided.

-In my opinion, besides being on a table, the drying methods should be described in the method sections and not merely shown in the results on a table format

Round 3

Reviewer 2 Report

Comments and Suggestions for Authors

I have no more comments